# The Association between Vaginal Dysbiosis and Reproductive Outcomes in Sub-Fertile Women Undergoing IVF-Treatment: A Systematic PRISMA Review and Meta-Analysis

**DOI:** 10.3390/pathogens10030295

**Published:** 2021-03-04

**Authors:** Axel Skafte-Holm, Peter Humaidan, Andrea Bernabeu, Belen Lledo, Jørgen Skov Jensen, Thor Haahr

**Affiliations:** 1Department of Clinical Medicine, Aarhus University, Denmark and the Fertility Clinic Skive, Skive Regional Hospital, 7800 Skive, Denmark; axel@mentor.dk (A.S.-H.); peter.humaidan@midt.rm.dk (P.H.); 2The Fertility Clinic, Instituto Bernabeu, 03016 Alicante, Spain; abernabeu@institutobernabeu.com (A.B.); blledo@institutobernabeu.com (B.L.); 3Statens Serum Institute, Research Unit for Reproductive Microbiology, 2300 Copenhagen, Denmark; JSJ@ssi.dk

**Keywords:** bacterial vaginosis, vaginal microbiota, IVF, next generation sequencing, qPCR, clinical pregnancy rate

## Abstract

Recent advances in molecular microbiology have enabled refined studies of the genital tract microbiota. This constitutes the basis of the present updated systematic review and meta-analysis which investigate vaginal dysbiosis (VD) as defined by either microscopy (e.g., Nugent score for bacterial vaginosis) or molecular methods (qPCR and Next Generation Sequencing) to evaluate the impact of VD on the reproductive outcomes in women undergoing IVF-treatment. A total of 17 studies were included, comprising 3543 patients and with a VD prevalence of 18% (95%CI 17–19). Across all methods, VD is a significant risk factor for early pregnancy loss in IVF (Relative risk (RR) = 1.71 95%CI 1.29–2.27). Moreover, a predefined sub-analysis of studies using molecular methods for VD diagnosis showed a significant reduction in the clinical pregnancy rate when compared to normal vaginal microbiota patients (RR = 0.55 95%CI 0.32–0.93). However, regardless of diagnostic methodology, VD did not significantly influence live birth rate (LBR). In conclusion, molecular tools have provided a more detailed insight into the vaginal microbiota, which may be the reason for the increased adverse effect estimates in IVF patients with molecularly defined VD. However, the quality of evidence was very low across all outcomes according to GRADE and thus, more studies are warranted to understand the impact of VD in IVF.

## 1. Introduction

Infertility is defined by an absolute failure to become pregnant, whereas subfertility refers to an inability of becoming pregnant after one year of unprotected intercourse without conceiving [1]. In this aspect, it is essential to medically examine the cause of infertility and subfertility, as it is important for fertility treatment and the outcome of assisted reproductive technology (ART). The most common reasons for infertility in the female are tubal factor, ovulation disorders and advanced female age [1,2]. Despite recent advances, numerous cases of subfertility, as well as reasons for early pregnancy losses, remain unexplained.

Whereas the vaginal microbiota and bacterial vaginosis (BV) for a long time have been investigated for preterm birth prevention [3], in recent years, the genital tract microbiota has also gained more attention in relation to early implantation, early pregnancy and live birth rate (LBR) [4,5]. Previously, it was assumed that the endometrium was a sterile cavity, but recent studies suggest the opposite—i.e., that bacteria are present in the endometrium [4,6]. This has led to the hypothesis that vaginal bacteria may ascend into the endometrium and, thus, affect reproduction [7]. Although the genital tract microbiota is a complex ecosystem of more than 200 different bacterial species [8], it is commonly accepted that a healthy genital tract microbiota is dominated by facultative anaerobic *Lactobacillus* spp., i.e., a *Lactobacillus* dominant (LD) microbiota [9]. In more detail, approximately 80% of asymptomatic reproductive age women have vaginal microbiota dominated by one of only four *Lactobacillus* (L.) spp.: L. *crispatus*, L. *gasseri*, L. *iners* and L. *jensenii* [10,11]. Interestingly, LD microbiota is more prevalent in pregnant women compared to non-pregnant women [12]. The reason for this is not entirely clear but could in part be explained by hormonal fluctuations influencing the vaginal microbiota. During pregnancy, elevated estrogen levels may increase the glycogen synthesis in the vaginal epithelium. Although *Lactobacillus* spp. do not utilize glycogen directly, the presence of the fermentation products has shown to support colonization of lactobacilli [13]. The physiological benefit of *Lactobacillus* spp. relates to the production of lactic acid which lowers the vaginal pH and protects against infection from pathogens [11,14], however, individual *Lactobacillus* spp. are not equally beneficial with regards to reproductive health [12,15].

BV is the most frequent VD reported in approximately 19% of infertile women [16]. BV is defined as an alteration of the vaginal microbiota, resulting in a heterogeneous dysbiotic vaginal environment with reduced concentrations of *Lactobacillus* spp. and an increased presence of typical BV bacteria such as *Gardnerella* spp., *Atopobium vaginae*, *and Mycoplasma hominis* [10,17,18]. Moreover, BV may involve the presence of a polymicrobial biofilm [17] strongly adhered to the vaginal epithelium [18]. Interestingly, up to 40 percent of BV-positive women display no symptoms [19]. BV is associated with implantation failure, early miscarriage [20,21] and preterm birth [15].

Many diagnostic approaches have been used in order to diagnose BV. The Amsel Criteria, consisting of the following findings: pH > 4.5, fishy odor, presence of Clue cells, and vaginal discharge [22] is the most commonly used clinical diagnostic method. Additionally, the laboratory gold standard Nugent microscopy scoring system has been used to evaluate the abundance of *Lactobacillus* spp. and BV-associated bacteria [23]. Based on studies using microscopy, Van Oostrum et al. [16] published the first meta-analysis (2013) on BV in relation to reproductive outcomes in IVF patients. Subsequently, two meta-analyses, one by our group [21] (2018) and one by Singer et al. [24] (2019) have been published. Although eligibility criteria for inclusion varied in previous meta-analyses, a correlation between BV and ART-failure, including lower rates of clinical pregnancy [21,24] and an association with early spontaneous abortion [16,21] has been found.

Through the last decade, studies have enabled a shift in the diagnostic methodology of the vaginal microbiota—from microscopy to molecular based approaches [25,26,27,28,29]. First, many groups validated qPCR diagnostic assays according to the Nugent score BV. Secondly, the introduction of next generation sequencing (NGS) has enabled taxonomic identification of a wide range of bacterial taxa using e.g., 16S rRNA gene sequencing. This has enabled the vaginal microbiota to be stratified into community state types [30,31] and some community state types might represent VD. Molecular methods for VD diagnosis are currently being studied in a clinical context, however, among many future research subjects, the diagnostic levels for vaginal bacterial abundance in relation to clinical outcome are not well-established and the impact of temporal changes in the vaginal microbiota remains unclear. The aim of this review was to evaluate the impact of VD on the reproductive outcomes of IVF patients, stratified by molecular and microscopy methods to investigate the current state of knowledge and to inform future research.

## 2. Results

The systematic literature search identified 108 citations. A total of 70 citations were excluded based on title and an additional 27 were excluded based on abstract. Hence, a total of 11 citations were assessed for eligibility by examination of the full text. Two studies were removed due to study design [32,33] and three were removed as it was not possible to extract pregnancy outcome data for meta-analysis [34,35,36]. Two studies were based on the same study population [5,29], and only the initial publication was included [5], but data on LBR was extracted from the later study [29]. A total of five studies [25,26,27,28,37] were included and added to the 12 studies [5,20,38,39,40,41,42,43,44,45,46,47] in the previous version of the present systematic review and meta-analysis [21]. Overall, the present meta-analysis included a total of 17 studies, comprising 3543 IVF patients. The full selection of the studies can be seen in the Preferred Reporting Items for Systematic Reviews and Meta-Analyses (PRISMA) [48], Flow Diagram, Figure 1.

### 2.1. Data Characteristics

A full view of individual study characteristics can be seen in Table 1. Overall, the prevalence of VD was 18% (95%CI 17–19) (644/3543) (Appendix A). However, a large heterogeneity of the VD prevalence was noticed in the studies, varying from 4% [47] to 44% [27].

Thirteen publications [5,20,37,38,39,40,41,42,43,44,45,46,47] were based on microscopy and resulted in an overall prevalence of 17% (517/3091). Two studies compared vaginal microbiota by Nugent Score with a qPCR defined abnormal vaginal microbiota (AVM), resulting in increased prevalence rates from 21 to 28% [5] and 6 to 9% [38], respectively. Kyono et al. [27] used 16S rRNA gene sequencing and found a prevalence of 44% while Vergaro et al. [28] used a qPCR-based method, reporting a prevalence of 23%. Both studies used <90% *Lactobacillus* spp. as criteria to diagnose VD. Koedooder et al. [26] used an IS-pro™ technique and reported a VD prevalence of 18%. This study defined an unfavorable microbiome profile by a relative *Lactobacillus* load <20%, relative load of L. *jensenii* >35%, presence of *Gardnerella vaginalis* or *Proteobacteria* >28% of total bacterial load. Finally, one study [25] used 16S rRNA gene sequencing without defining VD. As individual participant data was shared, identification of patients with VD was performed for the present meta-analysis according to the criteria defined by Koedooder et al. listed above. This resulted in a VD prevalence of 6% (2/31). Overall, molecular methods (qPCR, 16S rRNA and IS-pro) resulted in a prevalence of 19% (171/889). The prevalence ratio between microscopy and molecular methods was not significant, 0.87 (95%CI 0.74–1.02).

By evaluating the 17 studies it became evident that the timing of the sampling differed. Seven studies [20,38,41,42,45,46,47] performed sampling on the day of oocyte retrieval, while five studies [27,30,40,41,44] performed the sampling at the time of embryo transfer, revealing a BV prevalence of 16% (391/2446) and 17% (90/540) respectively. Three studies [5,28,45] performed vaginal microbiota sampling prior to IVF stimulation, presenting an overall BV prevalence of 24% (87/367). Finally, Kyono et al. compared a vaginal swab in different cycles and different menstrual phases [27], but no difference was observed. Moragianni et al. did not report time of sampling [37].

### 2.2. Live Birth Rate (LBR)

A total of 539 live births were recorded among 1699 patients (Appendix A). When comparing a VD to a normal vaginal microbiota in IVF-patients, data showed a relative risk (RR) of live birth per embryo transfer of 1.03 (95%CI 0.79–1.33; I^2^ = 28%). Subgroup analyses performed according to methodology showed a RR of 1.10 (95%CI 0.80–1.50; I^2^ = 32%) for microscopy while molecular methods revealed a RR of 0.80 (95%CI 0.47–1.35; I^2^ = 32%) (Table 2). The quality of evidence for live birth rate was very low according to The Grading of Recommendations Assessment, Development and Evaluation (GRADE) [49] (Appendix A).

### 2.3. Early Pregnancy Loss

A total of 235 early pregnancy losses were recorded among 1386 patients (Appendix A). The relative risk of early pregnancy loss per hCG positive pregnancy for VD patients undergoing IVF was 1.71 (95%CI 1.29–2.27; I^2^ = 0%) when compared to normal microbiota patients. A subgroup analysis of microscopy showed a RR of 1.61 (95%CI 1.17–2.20; I^2^ = 0%), compared to 2.12 (95%CI 0.91–4.90; I^2^ = 37%) when stratifying for molecular methods (Table 2). The quality of evidence on early pregnancy loss was very low according to GRADE [49] (Appendix A).

### 2.4. Clinical and Biochemical Pregnancy

A total of 1051 clinical pregnancies were recorded among 3315 patients (Appendix A). Overall results showed a RR of 0.84 (95%CI 0.68–1.04; I^2^ = 41%) per embryo transfer in clinical pregnancy rate when comparing patients with VD to patients with a normal microbiota. Subgroup analyses stratifying by molecular methods showed a significantly lower RR for clinical pregnancy per embryo transfer in VD patients, 0.55 (95%CI 0.32–0.93; I^2^ = 49%). In contrast, when stratifying for microscopy, RR was 0.95 (95%CI 0.78–1.16; I^2^ = 22%) for clinical pregnancy per embryo transfer.

By investigating biochemical pregnancy rate (serum HCG-positive) per embryo transfer it became evident, that no significant association was observed when comparing the two groups. An overall RR of 0.95 (95%CI 0.79–1.15; I^2^ = 38%) was found, while with microscopy a RR of 0.98 (95%CI 0.78–1.23; I^2^ = 47%) was found. Molecular methods showed a RR of 0.78 (95%CI 0.58–1.04; I^2^ = 0%) (Table 2). Overall, 1100 biochemical pregnancies were recorded among 2845 patients (Appendix A). The quality of evidence on clinical- and biochemical pregnancy was very low according to GRADE [49] (Appendix A).

## 3. Discussion

### 3.1. Main Findings

This systematic review and meta-analysis found a VD prevalence of 18% (95%CI 17–19) among infertile patients undergoing IVF-treatment, which is in line with previous reports [16,21]. Microscopy resulted in a VD prevalence of 17%, while molecular methods revealed a prevalence of 19%. Although this prevalence estimate could constitute a measure of VD in the general IVF population, we generally found that the prevalence of VD differed a lot among studies which call into question the general use of this estimate for specific populations and different VD definitions. Despite a significant increase in early pregnancy loss (RR = 1.71 95%CI 1.29–2.27), the overall meta-analysis indicated that VD did not significantly impact LBR, clinical pregnancy rates or biochemical pregnancy rates (Table 2). Interestingly, stratification for molecular diagnostic methods only, found a significant association between VD and the clinical pregnancy rate (RR = 0.55 95%CI: 0.32–0.93) (Table 2). This could be interpreted as an effect of a more accurate VD diagnosis using molecular based methods. Moreover, for all other reproductive outcomes investigated herein, the effect estimates were more pronounced when using molecular based methods as compared to microscopy, albeit not statistically significant. According to GRADE [49] the quality of evidence was very low across all outcomes. Thus, additional research is needed in order draw firm conclusions regarding VD in relation to reproductive outcomes in IVF patients.

### 3.2. Strengths and Limitations

A series of eligibility criteria were prepared in order to homogenize the studies included, however, despite this effort a widespread heterogeneity was observed, including age, ethnicity, diagnostic approach, and use of antibiotics (Table 1). Five studies [40,42,44,45,48] used antibiotics for either all patients or in patients with antibiotic-requiring diseases, which may have biased outcomes in the respective studies as well as in this meta-analysis. However, we previously analyzed the antibiotic impact on effect estimates [5,20,21,38,39,40,41,42,43,44,45,46,47] and as none of the added studies [25,26,27,28,37] reported use of antibiotics, we did not repeat this analysis in the present study. Most studies [26,27,28] did not share underlying individual participant data. Compared to the estimated prevalence of VD compiling microscopy and different molecular based VD diagnostic methods, individual patient data might have resulted in an optimization of the comparison between molecular and microscopy methods.

Furthermore, large inter-study differences in VD prevalence were observed, ranging between 4% and 44%. It became clear that sampling was performed at different time points during the menstrual cycle and in relation to IVF treatment (Table 1). This could have affected the prevalence of VD, as the vaginal microbiota may be influenced by hormonal fluctuations [50,51].

Another limitation to this study is the fact that molecular methods do not cover the vaginal mycobiome, which potentially could impact the association observed in this study. The most common yeast, *Candida albicans* share some of the same pathogenesis as *Gardnerella* spp. [52], which might affect reproduction.

Studies based on molecular methods in comparison to microscopy are few, only covering 889 patients compared to 3091 in studies based on microscopy. Unfortunately, the present molecular methods for VD diagnosis are not uniform by definition which may be a significant contribution to the interstudy heterogeneity observed in this meta-analysis. Overall, the strength of evidence was very low on all outcomes according to GRADE [49]. This finding add uncertainty to the estimates, which underline the importance of additional research.

The strength of this study, however, is the addition of five new studies, which enabled this meta-analysis to be the first to present a predefined sub-analysis, covering molecular methods for detection of VD and the risk of adverse reproductive outcomes in IVF patients. This review highlights important aspects in the transformation of the diagnostic methodology in relation to BV and provides a detailed summarization, across methods, for researchers in the field.

#### Interpretation

Overall, three systematic reviews [16,21,24] were identified in relation to this topic. Compared to the previous systematic review by our group [21], Singer et al. [24] included only six studies [5,38,39,40,41,42], concluding that a VD reduces the clinical pregnancy rate compared to patients with a normal vaginal microbiota (OR = 0.70 95%CI 0.49–0.99) [24]. We used RR for statistical analysis, reporting that VD did not significantly impact LBR, biochemical pregnancy rates, or clinical pregnancy rates, however a significant association with early spontaneous abortion (RR = 1.71 95%CI 1.29–2.27) was found in the overall analysis [21]. The present meta-analysis corroborates previous findings, as VD was associated with early pregnancy loss and a lower clinical pregnancy rate, yet the latter was only noticed in VD patients diagnosed by molecular methods. In our view, this is an important finding which may be due to a more accurate VD diagnosis using molecular methods.

In general, we observed a higher point prevalence of VD among molecular methods compared to microscopy. Two studies showed a numerical increase in the VD prevalence being 4% [38] and 7% [5] respectively both studies used a qPCR-method compared to Nugent Score. This increase was mainly caused by the dichotomization of the intermediate Nugent score patients, as previously described [21]. In contrast, two other studies [27,28] used the arbitrary cut-off level <90% *Lactobacillus* spp. as a definition of VD. Based on qPCR and relative cut-offs, Vergaro et al. reported no overall association between VD among IVF patients with blastocyst transfer who achieved a live birth compared to women with no live birth. However, a significantly higher LBR was noted among women whose vaginal microbiota was dominated by L. *crispatus* [28]. In support, Bernabeu et al. reported a significant association between *Lactobacillus*-dominated microbiota (mainly L. *crispatus*) in IVF patients achieving biochemical pregnancy, using 16S rRNA gene sequencing [25]. Moreover, based on the IS-pro™ technique Koedooder et al. noted, that a high abundance of *Lactobacillus* spp. positively impacted clinical pregnancy rates. However, in their study a relatively high abundance (>60%) of L. *crispatus* was shown to have a negative impact on pregnancy rates [26].

These equivocal findings indicate that studies [27,28] using an arbitrary cut-off level of <90% *Lactobacillus* spp., need to reconsider this cut-off, taking into account for example the total bacterial abundance and individual characteristics of different *Lactobacillus* spp. As an example, a recent study provided an excellent way to do this [53]. Interestingly, the relative abundance of vaginal *Lactobacillus* spp. is known to increase during pregnancy [54] potentially improving the development of a healthy pregnancy [25,26,27,40,46], albeit there is still no causal evidence to intervene on VD in pregnancy in order to improve reproductive outcome [3]. As stated earlier, *Lactobacillus* spp. protect against invasion and infection of opportunistic pathogens by lowering the pH in the vaginal tract [55]. The acidity depends on the *Lactobacillus* spp., as a lower pH was recorded in women with a high abundance of L. *crispatus* compared to L. *iners*, L. *jensenii* and L. *gasseri* [31]. This characteristic may prevent opportunistic pathogens from ascending into the endometrium and negatively affect reproduction. Moreover, this could in fact explain the higher abundance of *Lactobacillus* spp. in pregnant women compared to infertile women if the hypothesis of VD mediated infertility holds true.

One explanation for the differences in VD diagnosis across studies origins in the fundamental difference between NGS and qPCR. While qPCR is a well-established method to identify known species, NGS provides a more advanced taxonomic identification of nearly all bacteria in a distinct microbiota. NGS provides a relative abundance measure, which may underestimate the presence of some bacterial species, due to e.g., high total concentrations of other bacterial species. For example, L. *iners* has a 15-fold higher concentration than L. *crispatus* by qPCR [29], and another study using genus specific primers did not observe statistical difference in the total number of lactobacilli comparing normal vaginal microbiota to BV [56]. This could result in a microbiota, which by NGS would have a relative abundance skewed by a high total abundance of e.g., L. *iners* which may “cover-up” VD bacteria such as *Gardnerella* spp. In fact, it was recently shown that a high total abundance of *Gardnerella* spp. may be underestimated by relative 16S rRNA gene sequencing [53]. Typically, BV has a significantly higher total bacterial load compared to normal vaginal microbiota [56]. For this reason, it is important to acknowledge that underestimated bacteria by relative abundances may still have a physiological effect. In contrast, the drawback of qPCR which focuses on only a few bacterial species, may be that qPCR does not sufficiently cover the entire vaginal microbiota of the specific patient. A future combined approach, using both NGS and qPCR to identify potential bacterial pathogens might be the way forward. Recently, a network meta-analysis was published displaying a network map which provided an overview on the relationships of bacterial species and assessment on studies providing direct evidence for diverse vaginal microbiota compositions and certain outcomes [57]. For the future this approach could be considered as this would provide a more detailed overview for the reader. However, in order to conduct such a network meta-analysis, authors of future association studies are encouraged to share underlying individual participant data.

Unquestionably, molecular methods have provided a more detailed insight into the vaginal microbiota allowing researchers to understand its complexity. Molecular methods are not uniform, and this review has highlighted the variability in defining VD. The present meta-analysis showed that despite the variability, the molecular based VD was significantly associated with clinical pregnancy after IVF treatment. This could lead to more appropriate criteria for identifying women for future intervention studies which is needed to understand whether VD diagnosis and treatment should be encouraged in IVF patients.

## 4. Materials and Methods

The present PRISMA conforming meta-analysis systematically reviewed studies investigating the vaginal microbiota in relation to selected reproductive outcomes in IVF. The analysis is a pre-planned and updated analysis of a previously published meta-analysis [21] (2018) carrying the PROSPERO registration: CRD42016050603. The PRISMA Checklist and PRISMA Statement for Reporting Systematic Reviews [58] and the MOOSE guidelines for Meta-analysis in observational studies [59] have been used for quality assessment and can be found in Appendix A. Eligibility criteria for study inclusion in this analysis were predefined and correlated with the criteria published previously [21] (Table 3). The primary outcomes of this review were LBR and early pregnancy loss. Secondary outcome measures were clinical pregnancy rate (ultrasound verified heartbeat) and biochemical pregnancy rate (hCG serum-positive pregnancies). Many studies do not provide information about early pregnancy loss and, thus, clinical pregnancies were subtracted from biochemical pregnancies (Appendix A) in order to deduct the number of early pregnancy losses.

### 4.1. Literature Search Strategy

The PubMed (Medline) database was used to make an updated systematic literature search, using relevant keywords and MeSH terms (Appendix A). The previous systematic review [21] had 420 hits as of September 18, 2017. The present literature search found 528 publications as of September 27, 2020. Publications were screened by title, and subsequently by abstract by ASH and TH. If an abstract featured some of the eligibility criteria and/or outcomes, the publication was read in full. In cases of doubt, PH and JSJ were consulted to make a final decision on study inclusion. Additional searches in Embase, Scopus and Cochrane were conducted and generated no additional articles. For two studies, authors were contacted to provide additional data before study inclusion. One study [25] did not report individual microbiota-profiles allowing individual pregnancy outcome evaluation, but the authors who are co-authors of the present review provided relevant data for study inclusion. The authors of Moragianni et al. [37] were contacted to clarify their definition of pregnancy and the authors provided relevant data for study inclusion. Additionally, we contacted authors of three studies [34,35,36] but received no response and consequently their data could not be evaluated in the present meta-analysis.

### 4.2. Quality of Articles

In order to assess the quality of evidence included in the present systemic review and meta-analysis, the Newcastle-Ottawa Scale [60] was applied for each included study. Full publications were judged based on three parameters: selection of study group, comparability of groups and ascertainment of exposure or outcome. This scoring system helped identify bias and validation of the added articles [25,26,27,28,37]. Quality assessment can be found in Appendix A. Moreover, the quality of evidence was assessed using GRADE [49] for all outcomes. The basis for the evaluation can be found in Appendix A. Quality assessment of studies [5,20,38,39,40,41,42,43,44,45,46,47] included in the previous systematic review can be seen in that publication [21].

### 4.3. Data Extraction and VD Definition

Data extraction for individual studies included the following characteristics: author, analysis method, outcomes, and sample size of individual studies, and can be found in Appendix A. This meta-analysis chose to identify Nugent Score 0-6 as indicative of normal microbiota and 7–10 as indicative of BV/VD. Two studies [20,41] used modified Spiegel criteria [61] (MSC), which is very closely related to Nugent score. As with Nugent score, we chose to merge normal microbiota and intermediate microbiota as indicative of normal. Two studies [5,38] used both Nugent Score and qPCR to assess the vaginal microbiota. Overall results were based on qPCR data from these two studies, however subgroup-analysis featured data from both methods. Recent research has emended the description of *G. vaginalis* and made discovery of three new species; *G. leopoldii*, *G. piotii* and *G. swidsinskii* why this study refers to *Gardnerella* spp. as a group [62].

The qPCR diagnosis of VD was determined based on BV with a high total abundance of *Gardnerella* spp. and *A. vaginae* as first defined by Menard et al. in pregnant women [63,64]. This qPCR diagnosis was subsequently used in a modified version by Mangot-Bertrand et al. in IVF [38]. Later, the method was further refined by our group as based on Nugent score in IVF patients [5]. In contrast to the total abundance qPCR method mentioned above, studies by Kyono et al. [27] and Vergaro et al. [28] defined VD by a non-Lactobacillus dominated microbiota (NLD: <90% *Lactobacillus* spp.) and a normal microbiota defined by a Lactobacillus-dominated microbiota (LD: >90% *Lactobacillus* spp.), i.e., a relative abundance method. Finally, a recent study used the 16S–23S ribosomal operon interspace region to define VD in IVF [26], the so-called IS-pro™ technique. Based on 16S rRNA gene sequencing data, extradited from the study by Bernabeu et al. [25], the predictive cut-offs reported by Koedooder et al. [26] was used to define VD in that study.

Overall, this meta-analysis divided studies into two subgroups: Microscopy or molecular defined VD according to the criteria stated above. Microscopy comprised all studies based on Nugent criteria and MSC while the molecular group constituted qPCR, IS-pro™ technique and 16S rRNA gene sequencing.

### 4.4. Statistical Meta-Analysis

The overall effect, relative risk (RR) with a 95% confidence interval and Forest plot (Appendix A), were estimated in a random effects model, using Mantel-Haenszel method in REVIEW Manager version 5.3 (Cochrane, London, UK) [65]. VD prevalence’s, prevalence ratio, Funnel plots with pseudo 95% confidence limits and Egger’s test (Appendix A) were computed using Stata IC Version 16.1 (Statacorp LLC, Texas, USA). Egger’s test was performed to investigate potential publication bias in the GRADE [49] analysis (Appendix A). Patients lost to follow-up were excluded from analysis.

## 5. Conclusions

This systematic review and meta-analysis conclude that VD across all diagnostic methods is significantly associated with a higher early pregnancy loss rate among women undergoing IVF. In addition, stratification for molecular methods to diagnose VD revealed a significant negative impact on clinical pregnancy rates per embryo transfer. However, the quality of evidence was very low according to GRADE which warrants further research. Molecular methods have unquestionably provided a more detailed view of the vaginal microbiota and translate into more pronounced effect estimates for linking VD and reproductive outcomes in IVF patients. However, many research groups are working on various molecular diagnostics, in order to diagnose VD in IVF patients, which are not uniform. In addition, no diagnostic method has yet been proven to causally impact the reproductive outcome. Future association studies are encouraged to share underlying individual participant data in addition to the sequencing data allowing a more refined meta-analysis. This might lead to more precise evaluation of bacterial cut-off levels to diagnose VD and potentially create basis for a new gold standard using molecular methods to diagnose VD in IVF patients. Finally, future intervention trials of genital tract dysbiosis may be important to investigate causality and treatment strategies in infertility and IVF.

## Figures and Tables

**Figure 1 pathogens-10-00295-f001:**
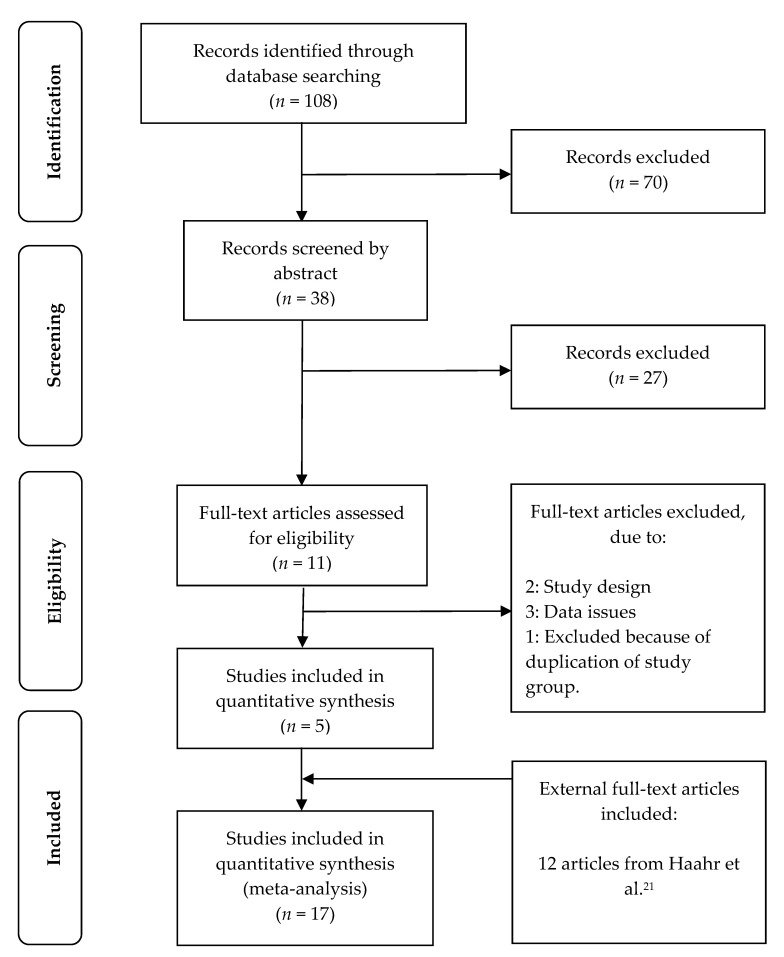
PRISMA Flow Diagram.

**Table 1 pathogens-10-00295-t001:** Study characteristics. Table legend: MSC = modified Spiegel criteria.

Study	Method	VD Prevalence	Age (Normal Microbiota)	Age BV	Antibiotics	Timing of Sampling	Country/Ethnicity	IVF Cycle of Sampling
Haahr et al. [5]	qPCR-Nugent	27.7%	31 (median)	30 (median)	No	2–4 weeks prior to IVF treatment. Maximum 2 months before transfer.	Denmark/90% Caucasian	Before IVF cycle sampling
Mangot-Bertrand et al. [38]	qPCR-Nugent score	9.4%	33.5 (mean)	33.9 (mean)	No	On the day of the oocyte retrieval	French/No data	4 cycles: T1, T2, T3 and ≥ T4
Selim et al. [39]	Nugent score	37%	21–44 (range) for all included patients		Metronidazole at oocyte retrieval (twice daily for five days)	At the time of embryo transfer	Egypt/No data	First cycle
Eckert et al. [40]	Nugent score	11%	21–45 (range) for all included patients		Doxycycline (100 mg orally, twice daily for 5 days) at oocyte retrieval.	At the time of embryo transfer	USA (Washington)/No data	First cycle
Liversedge et al. [41]	MSC	25.6%	33.4 (median)	33.6 (median)	Positive *C. trachomatis* serology was treated with ofloxacin.	On the day of the oocyte retrieval	England/No data	Any cycle
Gaudoin et al. [42]	Nugent score	16.3%	No data		No data	On the day of the oocyte retrieval	Scotland/No data	No data on cycle
Boomsma et al. [43]	Nugent score	8.6%	34.8 (mean)	36.7 (mean)	Endometriosis or tubal pathology (17%) received a single dose of antibiotics (Ampicillin+clavulanic acid and doxycycline) before oocyte retrieval.	At the time of embryo transfer	The Netherlands/No data	No data on cycle
Eldivan et al. [44]	Nugent score	37.8%	31 (mean) for all included patients		BV+: Metronidazole 500mg oral x2 for 7 days + metronidazole intravaginally. Azitromycin 1g was given to *Chlamydia* positives	Specimens were collected immediately after menses	Turkey/No data	No data on cycle
Moini et al. [45]	Nugent score	7.3%	28.6 (mean)	28.3 (mean)	No data	On the day of the oocyte retrieval	Iran/No data	No data on cycle
Moore et al. [46]	Nugent score	13.2%	21–45 (range) for all included patients		Doxycycline treatment was started after egg retrieval for 5 days.	Vaginal swab at oocyte retrieval and embryo transfer	USA (Washington)/No data	No data on cycle. Only one cycle per patient was included, although it was not necessarily the subject’s first IVF cycle.
Ralph et al. [20]	MSC	24.6%	33 (median) for all included patients		No data	On the day of the oocyte retrieval	England/95% Caucasian.	No data on cycle
Spandorfer et al. [47]	Nugent score	4.23%	No data		All patients: tetracycline and methylprednisolone at oocyte retrieval and for four days.	On the day of the oocyte retrieval	USA/No data	No data on cycle
Moragianni et al. [37]	Nugent score	36.9%	32 (median) for all included patients		No data	No data	Greece	No data on cycle
Vergaro et al. [28]	qPCR	23.3%	41.2	42.3	No data	At the time of embryo transfer	Spain/100% Caucasian	Donated oocytes (no data on cycle)
Koedooder et al. [26]	IS-pro™	17.7%	20–44 (range) for all included patients		No data	Within 2 months prior to ET: self-collected vaginal swab + midstream urine sample before IVF or IVF-ICSI start.	The Netherlands/No data	No data on cycle
Kyono et al. [27]	16S rRNA	44.3%	37 (mean) for all included patients		No data	Vaginal swab: collected in different cycles and different menstrual phases.	Japan/100% Japanese	Follicular phase, Ovulation phase, Luteal phase
Bernabeu et al. [25]	16S rRNA	6.5%	40 (median) for all included patients		No data	At the time of embryo transfer	Spain/100% Caucasians	All cycles were of frozen embryo transfers under artificial endometrium preparation

**Table 2 pathogens-10-00295-t002:** Relative risk on reproductive outcomes.

Outcome	RR (CI 95%)	No. Of Participants (Studies)	Quality of Evidence (GRADE)	Reference & Comments
Primary outcomes				
**Live birth rate**	**1.03** (0.79–1.33)	1699 (9 studies)	⊕⊖⊖⊖ Very low *	See Appendix A.
Microscopy	**1.10** (0.80–1.50)	1231 (6 studies)	⊕⊝⊝⊝ Very low *	See Appendix A
Molecular	**0.80** (0.47–1.35)	543 (4 Studies)	⊕⊝⊝⊝ Very low *	See Appendix A
**Early pregnancy loss**	**1.71** (1.29–2.27)	1386 (14 studies)	⊕⊝⊝⊝ Very low *	See Appendix A.
Microscopy	**1.61** (1.17–2.20)	1179 (11 studies)	⊕⊝⊝⊝ Very low *	See Appendix A.
Molecular	**2.12** (0.91–4.90)	245 (4 studies)	⊕⊝⊝⊝ Very low *	See Appendix A.
Secondary outcomes				
**Clinical pregnancy rate**	**0.84** (0.68–1.04)	3315 (17 studies)	⊕⊝⊝⊝ Very low *	See Appendix A. From Moore et al. [46] we used LBR.
Microscopy	**0.95** (0.78–1.16)	2573 (12 studies)	⊕⊝⊝⊝ Very low *	See Appendix A.
Molecular	**0.55** (0.32–0.93)	826 (6 studies)	⊕⊝⊝⊝ Very low *	See Appendix A.
**Biochemical pregnancy rate**	**0.95** (0.79–1.15)	2845 (14 studies)	⊕⊝⊝⊝ Very low *	See Appendix A. From Moore et al. [46] we used LBR.
Microscopy	**0.98** (0.78–1.23)	2374 (11 studies)	⊕⊝⊝⊝ Very low *	See Appendix A.
Molecular	**0.78** (0.58–1.04)	555 (4 studies)	⊕⊝⊝⊝ Very low *	See Appendix A

* Symbols according to GRADE [49]. For quality assessment see Appendix A.

**Table 3 pathogens-10-00295-t003:** Eligibility criteria.

Eligibility Criteria
Population: Infertile woman attending IVF-treatment, all causes. Following methods were emphasized, when investigating on the vaginal microbiota (exposure): Microscopy, PCR technology, Fluorescence In Situ Hybridization (FISH) and Next Generation Sequencing. Outcomes: Studies reporting on one of the critical or important outcomes. Human studies only. Primary research article only. English language only. Cut off year 1980 Sub-Saharan African studies were excluded due to a higher background prevalence of competitive co-infections with BV, e.g., HIV, *Trichomonas vaginalis* and *Chlamydia trachomatis*. Case reports and Reviews were excluded.

## Data Availability

The data presented in this study are available in Appendix A.

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
