# Peer review of "The Association between Vaginal Dysbiosis and Reproductive Outcomes in Sub-Fertile Women Undergoing IVF-Treatment: A Systematic PRISMA Review and Meta-Analysis"

_pathogens, 2021, doi:10.3390/pathogens10030295_

Round 1

Reviewer 1 Report

The manuscript by Skafte-Holm et al is based on a previous meta-analysis performed by the senior author of the study. This previous work has shown very similar results, even though studies based on molecular methods as qPCR and NGS are expected to be an advancement compared to diagnosis of vaginal dysbiosis based on microscopy.

As such and without a more detailed evaluation of the bacterial strain resolution used, the study is unfortunately not adding much information to the field.

Detailed comments:

Introduction

The introduction is lacking information on the current knowledge on the composition of a normal vaginal microbiome and this needs to be added. A vaginal microbiome with microbial communities dominated by species of Lactobacillus has been associated with health, whereas a microbiome dominated by complex microbial communities of Gardnerella, Atopobium, Dialister, Peptoniphilus, Lachnospiraceae members (such as BVAB1) and other anaerobeshas been associated with dysbiosis and health risks. A complex vaginal microbiome is associated with bacterial vaginosis but there are still many women with vaginal microbiome compositions lacking Lactobacillusspecies that do not show symptoms.

At this point, it is extremely difficult to define vaginal dysbiosis based on molecular methods because of the lack of knowledge on the dynamic changes of the vaginal microbiome during the menstrual cycle and agreed cut offs for Lactobacillusspecies. This lack of knowledge needs to be mentioned here.

Line 44: change believed to assumed (scientists don’t believe, they prove hypotheses)

Line 47-53: refer to landmark publications in the field of the vaginal microbiome such as the work of the Human Microbiome Project (Serrano MG et al, Nature Medicine 2019)

Line 58: reference 19 is not a relevant reference, change

Results

Line 84: explain the abbreviation PRISMA

Figure 1

The arrow-head for the 12 articles included from Haahr et al needs to point to the opposite direction

Strenghts and limitations

This part needs to be improved to provide the reader with a better picture of the value of this metaanalysis. There are very few studies using molecular methods included and each of them uses different analysis technologies that cannot be compared directly. For the human vaginal microbiome, distinguishing between different Lactobacillusspecies is crucial, since e.g. Lactobacilluscrispatusseems to play a protective role not exerted by Lactobacillusiners. 16S rRNA gene sequencing is generally regarded to provide taxonomic resolution down to the genus level only. Furthermore, clinically important species such as Mycoplasma genitaliumand Chlamydia trachomatishave unusual substitutions in their rRNA genes and are often missed using 16s rRNA gene sequencing.

Another important limitation is the lack of data on when samples are taken in relation to the menstrual cycle as this is expected to influence the composition of the vaginal microbiome. Also the inconsistencies regarding the sampling in relation to embryo transfer.

None of the molecular methods covers fungi – discuss if there is an expected role of fungi.

Literature search strategy

Why is the study based on Pubmed only? Meta-analysis needs to cover all available data sources and the authors are requested to search Embase, Cochrane database and Web of Science.

Interpretation

There is an obvious limitation of the study because of the different molecular diagnostic methods used and this is a such an important question that the validity of the study can be questioned. Recently, network meta-analysis studies have been published where a network map was constructed to provide an overview of the network relationships of bacterial species and to assess how many studies provided direct evidence for the different vaginal microbiota compositions and certain outcomes (egg. Norenhag j, 2020, BJOG). For the future, this approach should be considered and mentioned in the discussion.  

I do miss a discussion on how the improved diagnostic tools can be used in a clinical setting.

Conclusions

Please reframe, different methods are not uniform by definition, what is your message? Furthermore, a diagnostic method will never causally impact biology, it is purely a tool to identify cases at risk.

Reviewer 2 Report

The study “The association between vaginal dysbiosis and reproductive outcomes in sub-fertile women undergoing IVF-treatment: a systematic PRISMA review and meta-analysis” by Skafte-Holm et al., has some strengths and limitations. This study is timely. However, the manuscript needs some clarifications. In the comments to the authors, do have some suggestions for its improvement. 

  • G. vaginalis was the only recognized species in its genus for four decades, but recently an emended description of G. vaginalis and descriptions of three new species – Gardnerella leopoldii, Gardnerella piotii, and Gardnerella swidsinskii – have been proposed. As such the authors needs to change the manuscript “G. vaginalis” to Gardnerella spp.

  • Line 57. Prevotella bivia needs to be included. This bacterial species has been associated with incident BV (DOI: 1093/infdis/jiy243). As such, this bacteria species should be also addressed in all the analyses. Furthermore, the fact that BV is a vaginal infection characterized by a presence of a polymicrobial biofilm should be also addressed in the introduction (doi: 10.1093/femsre/fuz027).

  • The PNA-FISH method to diagnose BV should be also included in the introduction section (doi: 10.7717/peerj.780).

  • Line 76-84 needs to be improved to be easier to follow the Figure 1.

  • There are other studies (not included in the manuscript) that compare the Nugent criteria with PCR. The authors should justify why these types of studies are not eligible to include in the analysis. Such articles did not follow the Newcastle-Ottawa Scale (quality parameters).

  • In the supplementary material file the authors should change the term “flora” to microbiota. The term flora is more appropriate to use for the Kingdom of plants.

Reviewer 3 Report

In this study the authors updated a previous systematic review and meta-analysis (SRMA) published in 2018. The aim of these reviews was to evaluate the impact of vaginal dysbiosis (VD) on the IVF outcomes. Unfortunately, both the reviews did not provide conclusive results. The main issue with the present review is the statistical analysis, but there are also other points that need to be addressed.

1.In the previous SRMA the Authors adhered to the PRISMA and MOOSE reporting guidelines. We think that in this new SRMA they should also follow both the guidelines.

2.The PRISMA and MOOSE checklists should be provided in the supplementary material. 3.In the previous SRMA the Authors applied the GRADE quality of evidence assessment. We suggest using the same methodology to the present SRMA.

4.Forest plot for each outcome should be included in the manuscript or in the supplementary material.

5.P-value of the heterogeneity test should be provided.

6.Potential publication bias should be assessed using the Egger's test and the creation of funnel plots for visual inspection (as the Authors did in their previous SRMA).

7.A full report of the results of the subgroup analysis should be shown in Table 3

Minor comments:

1.The Authors should explain why only one database was searched

2.A fourth systematic review should be considered (https://doi.org/10.1111/aji.13037)

3.It should be considered that the vaginal microbiome has been categorized into five groups (https://doi.org/10.1073/pnas.1002611107). Moreover, several studies indicate that the different species of Lactobacillus may have a different impact on fertility and IVF outcomes (https://dx.doi.org/10.3390). This information should be discussed in the Introduction.

4.In a previous paper the Authors showed that the predictive value of 16S ribosomal RNA gene sequencing was not superior to the simpler and less expensive qPCR diagnostic approach in predicting the risk of a poor reproductive outcome in patients undergoing IVF. https://doi.org/10.1093/infdis/jiz637). This could be of interest to the reader and should be included in the manuscript.

Round 2

Reviewer 2 Report

The authors have been very thorough in their response to the comments made, and have added additional information that improves the overall quality of the work. I have no further comments.

Author Response

Dear Reviewer,

Thank you for your kind reply and once again for reviewing the manuscript.

Reviewer 3 Report

We sincerely congratulate the Authors for their huge efforts in responding to the reviewer’s comments. As consequence the manuscript was significantly improved, and now we are confident that it will be very interesting for the readers. 

There is only one issue that was not addressed which is relevant and should be discussed. We have suggested to consider that the different species of Lactobacillus may have a different impact on fertility and IVF outcomes (see Campisciano et al. Microorganisms 2021, 9, 39. doi.org/10.3390/microorganisms9010039;  Štšepetova et al., Reprod Biol Endocrinol, 2020. 18(1): p. 3. Manuscript ref. #37). This could explain some conflicting results across the studies included in the SRMA and it should be taken into account in the manuscript.

Author Response

Dear Reviewer,

First of all, we would like to thank your kind reply and once again for reviewing the manuscript. 

Unfortunately, in our point-by-point response to your initial comments under ‘minor comments’ point 3, manuscript changes were left out. Additional information regarding the topic was already included in the Introduction Section page 2, line 58-59, as requested in the first review.

Also, we would like to mention that additional information in relation to your request can be found in the Discussion Section page 9, line 119-122

We hope you find these corrections sufficient.